# Secure Combination of IoT and Blockchain by Physically Binding IoT Devices to Smart Non-Fungible Tokens Using PUFs

**DOI:** 10.3390/s21093119

**Published:** 2021-04-30

**Authors:** Javier Arcenegui, Rosario Arjona, Roberto Román, Iluminada Baturone

**Affiliations:** Microelectronics Institute of Seville (IMSE-CNM), University of Seville, CSIC, C/Américo Vespucio 28, 41092 Seville, Spain; arjona@imse-cnm.csic.es (R.A.); roman@imse-cnm.csic.es (R.R.); lumi@imse-cnm.csic.es (I.B.)

**Keywords:** IoT security, blockchain technology, Ethereum, smart contracts, non-fungible tokens (NFTs), physical unclonable functions (PUFs), trusted hardware, secure boot

## Abstract

Non-fungible tokens (NFTs) are widely used in blockchain to represent unique and non-interchangeable assets. Current NFTs allow representing assets by a unique identifier, as a possession of an owner. The novelty introduced in this paper is the proposal of smart NFTs to represent IoT devices, which are physical smart assets. Hence, they are also identified as the utility of a user, they have a blockchain account (BCA) address to participate actively in the blockchain transactions, they can establish secure communication channels with owners and users, and they operate dynamically with several modes associated with their token states. A smart NFT is physically bound to its IoT device thanks to the use of a physical unclonable function (PUF) that allows recovering its private key and, then, its BCA address. The link between tokens and devices is difficult to break and can be traced during their lifetime, because devices execute a secure boot and carry out mutual authentication processes with new owners and users that could add new software. Hence, devices prove their trusted hardware and software. A whole demonstration of the proposal developed with ESP32-based IoT devices and Ethereum blockchain is presented, using the SRAM of the ESP32 microcontroller as the PUF.

## 1. Introduction

The Internet of things (IoT) is the paradigm where anything (devices, objects, subjects, etc.) can be interconnected through the Internet with the capability of interacting, collecting, processing, and sharing data in a smart way. The rapid growth in the number of things connected to Internet has led to the need for security solutions. Most existing solutions focus on the secure handling and sharing of the data, addressing risks that range from the use of personal data without the owner’s consent or knowledge to data access or manipulation by unauthorized parties [1]. Blockchain technologies have recently been used with IoT since they provide a distributed and cryptographically secure chain of data blocks, which allows an immutable data trail that guarantees ownership of data and user privacy [2]. Each block in a blockchain is identified univocally and is linked to the previous one by using a cryptographic hash function. A new block is added to the chain if participants in the blockchain with the role of miners demonstrate that the new block is secure and most of the miners agree with the demonstration, applying a consensus algorithm. If an attacker tries to change data in a block, for example, data captured by the sensors of an IoT device, the hash of the block changes and the data in the next block do not match. The attacker would have to change all the subsequent blocks in the chain to not be detected [3].

The data in the blocks can be data captured by the IoT devices, as well as data about transactions and the participants involved in the transactions. The exchange of digital currency or generic assets usually involves transactions between blockchain participants. In recent public blockchains such as Ethereum, agreements between participants are formalized through smart contracts represented by scripts. These scripts are validated as part of transactions by the consensus algorithm and, thus, once validated, the smart contracts are tamper-proof like the other data registered in the blockchain [4]. In addition, protocols are applied to achieve secure execution of smart contracts [5]. All participants execute the code of a smart contract in the same way and can check if the obligations established by the contract are met. Therefore, in addition to integrity, blockchain allows transparency because any participant can consult the data registered, including the smart contracts. This has been exploited to integrate blockchain with supply chain management, which, in the case of the IoT ecosystem, allows guaranteeing the origin and trustworthiness of data without the need for intermediaries [6]. The IoT devices can be fully autonomous to directly trade their data with third parties through smart contracts that govern all policies of data trade and ownership rights [7].

Ownership information can be included in smart contracts by means of tokens. Since a token is the digital representation of an asset in the blockchain, there are two main types of tokens, fungible and non-fungible tokens (NFTs), depending on the represented asset. Fungible tokens are identical and interchangeable tokens (such as a fiat currency), which allow accounting and billing transactions. Non-fungible tokens are unique and non-interchangeable tokens (such as notarial instruments or artwork collectables), which allow traceability of unique physical or non-physical possessions. An example of a non-physical possession in the context of IoT is the capability to access to resources or services. An IoT device acting as provider can represent by NFTs the resources or services provided. If it grants one of these NFTs to an IoT device acting as client, the client device is authorized to access the unique resource or service represented by the NFT [8,9]. Another wide use of NFTs is to represent physical products. Each product manufactured by a cloud service can be represented by a NFT such that its price, provenance, and ownership can be traced by the blockchain [10]. Applying a finer tokenization, each component of a product can be represented by a NFT such that the creation of a product from its components is registered in the blockchain as the minting of a new token from the tokens representing its components. In this way, not only the origin but also the subsequent transformation of a product can be traced [11].

This paper focuses on exploiting NFTs to make IoT devices secure, from a hardware and software point of view, such that the data they provide become trusted. The blockchain technology is exploited not only to ensure the trustworthiness of the hardware device manufacturing, but also to guarantee the security of its software. Otherwise, all security fails [12]. Concerning the software, special attention is given to the secure boot of the microcontroller in the IoT device hardware, whose objective is to check that the code to be executed is the expected one, and to the secure communication with owners and users, which can change device software. Ethereum blockchain is considered since it is one of the most extended public blockchains.

In the Ethereum community, which develops ERC (Ethereum Request for Comments) standards [13], the ERC-20 standard defines fungible tokens and the ERC-721 standard defines NFTs. An important attribute of an ERC-721 NFT is its owner, which is identified uniquely by their blockchain account (BCA) address. A blockchain participant interacts with the blockchain through a BCA, which is composed of a pair of cryptographic private and public keys and a unique address derived from the public key. In addition to an owner, a unique identifier is associated with an ERC-721 NFT. In the proposals where the ERC-721 tokens represent physical goods, the token identifiers are usually related to the logical attributes of the product, such as its bar or QR code [10], or to the product information data, such as disposal instructions and expiration dates [11]. The problem is that these identifiers are not related to something intrinsic of the product, i.e., they are not true product identifiers since they can be modified, copied or transferred to another product. To avoid this problem, crypto anchors were proposed as a tight link between the digital and physical domains [14]. Silicon physical unclonable functions (PUFs), which exploit the variabilities of the semiconductor manufacturing process [15], can be treated as crypto anchors for electronic products such as IoT devices. The binding of ERC-721 NFTs with physical IoT devices using PUFs was proposed in [16]. That work included, as a new attribute in ERC-721 NFTs, the blockchain account (BCA) address associated with the IoT device. The use of PUFs in the hardware of the IoT device was proposed to reconstruct the private key from which the BCA address of the device in its associated token is derived. In addition, that work also included, as a new attribute in ERC-721 NFTs, the blockchain account (BCA) address associated with the user of the IoT device, such that not only the ownership but also the use of the device can be traced by the blockchain with the same token. However, that work did not allow detecting if the device is not operating correctly, if the link between the device and the NFT is broken, or if the engagement with owner and user is lost at some time. To avoid these security flaws, this paper introduces a new NFT, named smart NFT, which strengthens the link between the IoT device and the NFT used to represent it.

The main contributions of this paper are the following:The proposal of smart NFTs to represent IoT devices that can participate actively in the blockchain by its BCA (to receive and provide information, and to sign transactions) and that can operate in several modes. The attributes defined for smart NFTs are not only the owner BCA address (as in the ERC-721 standard), the user BCA address, and the device BCA address (as in [16]). In addition, token states are included to trace the operating modes of the IoT device, public data are added to trace if secure communication channels are established between a device and its owner and user, and timestamps and timeouts are added to register the last validation of the link between the token and the device.A solution that allows IoT devices proving during their lifetime that their hardware and software are trusted because they are able to be bound to their smart NFTs and operate as expected. Physical attacks, which are common at IoT devices, e.g., their replacement by counterfeit devices or modifications of the content of their non-volatile memory, are detected after a controlled delay time thanks to the use of PUFs and the execution of a secure boot process.A solution that allows registering in the blockchain whenever shared secrets are agreed between devices and owners and between devices and users. From them, fresh session cryptographic keys can be derived for secure communication. Therefore, the trustworthiness of the devices can be traced even if there is a change of owners and users that manage them.A whole demonstration of the proposal developed with ESP32-based IoT devices and Ethereum blockchain. In this demonstration, the internal SRAM of the ESP32 microcontroller acts as the PUF and a true random number generator (TRNG), and it is controlled by a firmware developed in ESP-IDF (Espressif IoT Development Framework) [17]. The smart contract was developed with Remix [18], and the DApp (Decentralized Application) interfaces were created for the roles of manufacturer, owner, and user of the devices. These DApps were connected with the blockchain using a web interface and Metamask [19], and with the device using UART (Universal Asynchronous Transmitter Receiver) serial communication.

The rest of the paper is structured as follows: Section 2 reviews the related work on PUFs and NFTs associated with IoT devices. Section 3 describes the smart non-fungible tokens proposed to represent IoT devices, since the related work of IoT paradigm requires an extension of the ERC-721 tokens. The way of physically binding IoT devices to smart NFTs using PUFs, as well as the preservation of the bounding by executing trusted software are described in Section 4. The whole demonstration of the proposal is presented in Section 5. Lastly, conclusions are given in Section 6.

## 2. Related Work

A usual way to assign a cryptographic identity to a device is to assign it a private cryptographic key. Since secure non-volatile memories are expensive, a simple way of storing private keys in low-cost IoT devices is the use of internal eFuses or one-time programmable memories. However, this solution does not maintain private keys as secret but only provides data integrity, i.e., its contents cannot be modified but can be read. In fact, in the case of IoT devices using old versions of ESP32 microcontrollers, it was shown in [20] that, by injecting voltage glitches into the microcontroller, the protection of the keys could be bypassed and intended secret data stored in eFuses could be read.

A more recent and low-cost solution to identify devices is the use of PUFs. The blockchain-based proposals in the literature that consider PUFs to identify IoT devices do not employ NFTs explicitly. In [21], the challenge–response pairs provided by PUFs were employed for counterfeit detection of components of IoT devices. The manufacturer, using the generated PUF-based unique identifier, registers each device component in the blockchain with a smart contract. PUF-based unique identifiers and ownership transfer records were also employed in [22,23] to establish integrated circuit traceability. In [24,25], instead of storing the PUF response directly in the blockchain, the registered manufacturers store a cryptographic hash of the device identifier so that any end-user can verify the authenticity of the device if its associated hash is present in the blockchain. 

Theoretically, PUFs produce the same response with high probability. However, PUF responses can be affected by system noise and environmental conditions (such as temperature and voltage variations). Hence, cryptographic hash functions cannot be applied directly to PUF responses to obtain a reliable identifier. The ISO/IEC DIS 20897 [26] specifies the methods that can be employed to alleviate noise and generate non-stored cryptographic parameters from PUFs. Usually, fuzzy extractors apply error correcting codes to PUF responses to obfuscate and recover cryptographic keys without bit errors. Fuzzy extractors are employed in authentication protocols that verify that only the authentic device is able to recover the cryptographic key from the obfuscated stored data. In this way, in [27,28], the authentication procedure of an IoT device was based on checking if the cryptographic key is correctly recovered by the PUF. In [27], optical PUFs were employed in private blockchains. In [28], SRAM PUFs inside microcontroller-based devices were employed in zero-knowledge protocols. Only the public keys generated by the devices are stored in the blockchain. The blockchain simplifies the management of the device public keys and reduces the complexity of maintaining a central database.

Instead of PUFs, hardware security modules (HSMs) were employed in [29] to generate cryptographic keys for IoT devices. In [30], an HSM was included in each vehicle of a blockchain-based vehicle sharing platform. The HSM is utilized to generate randomly a pair of a public and a private key. The private key is used to sign transactions and the BCA address of the vehicle is derived from the public key. Two ERC-721 tokens are employed, one of them is owned by the vehicle and the other is owned by the user of the vehicle. However, an onboard HSM is more costly and complex for an IoT device than an intrinsic silicon PUF. In addition, an HSM does not generate a physical link with the device. Hence, we prefer the use of PUFs to establish a low-cost root of trust from which a private key and then the BCA address of the IoT device can be generated, as in [16]. Instead of using several ERC-721 tokens, as in [29], we prefer to introduce smart tokens, which extend the ERC-721 tokens and allow using fewer tokens in the applications.

Different Ethereum improvement proposals (EIPs) have extended the ERC-721 NFTs to allow for more versatility in certain use cases [13]. An example is the ERC-1155 Multi Token Standard, which allows combining fungible and non-fungible tokens in the same token. Other proposals in draft status are the EIP-2981, proposed to handle royalty payments, and the EIP-2615, to support mortgage and rental functions. The draft EIP-1948 introduces a non-fungible token that has dynamic data, that is, data that can change during the lifetime of the token. However, none of these extensions of ERC-721 tokens are addressed for IoT devices that can interact actively with the blockchain. The extension of the ERC-721 token proposed in [16] is not enough to detect when and why the link between a device and its token is broken. In addition, the transference of the token to a new owner is done automatically without registering the smart contract for any mutual authentication between the new owner and device. Hence, malicious owners that transfer non-operative devices cannot be proven. Lastly, the transference of the token to a new user is completed with the approval of the user but without registering the smart contract for any mutual authentication between the new user and device. Hence, malicious users that misuse devices cannot be proven. To represent IoT devices more securely, the next section describes our proposal of smart non-fungible tokens.

## 3. Proposal of Smart Non-Fungible Tokens (NFTs) to Represent IoT Devices

The ERC-721 standard [31] defines how to build NFTs on the Ethereum blockchain through a smart contract interface. Since the ERC-721 standard regards NFTs as unique assets that can be owned and transferred, it defines two main attributes: *tokenId*, which is the unique identifier of the token, and *owner*, which is the blockchain account (BCA) address that owns the token, can transfer ownership, and can approve others (*approved* and *operators*) to act in its name. However, something to consider when trying to represent IoT devices by NFTs is that IoT devices cannot be considered as passive assets with unique identifiers and owners; they need more attributes and functions to define them. Hence, an extension of the ERC-721 standard is presented below.

### 3.1. Attributes of Smart NFTs

In the context of a blockchain, a main difference between an IoT device and a passive asset is that the IoT device can interact actively with other blockchain participants, i.e., it can have a BCA. Hence, we propose the addition of the NFT attribute *device*, which is the BCA address associated with the IoT device. Furthermore, an IoT device is not only a possession of an owner but also an active agent that obeys a user to carry out certain tasks in an application. Hence, we propose the addition of the NFT attribute *user*, which is the BCA address of the user of the IoT device. In order to represent the dynamic activity of the IoT device, six more attributes are proposed: *timestamp*, *timeout*, *state*, *hashK_OD*, *hashK_UD*, and *dataEngagement*. The first of them, *timestamp*, registers in the blockchain whenever the device checks it is bound with its token. This is very important to register if the link is alive or not. The second of them, *timeout*, is the maximum delay time established for the device to prove again the bounding. If it is exceeded, the device is considered to be malfunctioning. The rest of the attributes are explained below.

An IoT device is a dynamic asset that can change of operating modes, which can be represented by states. In particular, the states defining if the device has been engaged or not with an owner or with a user deserve attention because the software to be executed by the device, which should be verified prior to be executed, is different. Hence, we propose the addition of the NFT attribute *state* and propose that the operating mode of the device should be in correspondence with its token state. We consider four main states of the token. The state *Waiting for owner* defines the situation whenever the token is created or transferred to a new owner, but the device and owner have not yet been mutually verified. Once they verify each other, the device recognizes its owner and the state of its token changes to *Engaged with owner*. In this state, the owner can transfer the token to a user. If the token is transferred to a user that coincides with the owner, the token state changes to *Engaged with user* and the device is ready to operate in its application, obeying the commands from its user. Otherwise, the token changes its state to *Waiting for user*. Once the user and the IoT device are verified mutually, the token changes its state to *Engaged with user*. From this state, the token can be transferred to another user, thus returning to the *Waiting for user* state. To cope with a user that cannot reply, the token can be transferred to another user if its state is *Waiting for user*. From all the states, the token can be transferred to another owner, thus returning to the *Waiting for owner* state. Figure 1 illustrates the state diagram of the token. The state changes are controlled by token functions as explained in the next subsection. 

The engagements of the device with an owner and with a user are carried out after mutual authentication protocols based on elliptic curve Diffie–Hellman key exchange protocols. These protocols allow a key agreement between the device and its owner, on the one hand, and the device and its user, on the other hand. Since the establishment of a shared secret is very important for a secure communication between them, we propose the inclusion of the attributes *hashK_OD*, *hashK_UD*, and *dataEngagement*. The first two attributes define, respectively, the hash of the secret shared between the device and its owner and that between the device and its user. Devices, owners, and users should check whether they are using the correct shared secrets. The attribute *dataEngagement* defines the public data needed for the agreement. If the mutual authentication fails, *dataEngagement* allows detecting which parts failed. This is more thoroughly explained in the next section. 

Table 1 shows the attributes of smart NFTs. The standard attributes *approved* and *operator* (which help the *owner* to transfer ERC-721 NFTs to other owners) are omitted in Table 1 because they are not in the scope of this work. Of course, they can also be considered in the proposed smart NFTs.

### 3.2. Functions and Events of Smart NFTs 

Table 2 shows the main functions and events of a smart NFT, which are defined in its interface. The pseudo-codes of the functions *createToken*, *updateTimestamp, setTimeout*, and *checkTimeout* are shown in Table 3. The function *createToken* is implicit in the ERC-721 standard. However, the proposed smart NFT defines it explicitly to link the device BCA address to the tokenId. In our proposal, only an agent identified as *manufacturer* can create the token. It is defined in the constructor of the smart contract associated with the smart NFT. The function *updateTimestamp*, which can be carried out only by the device, can be seen as a proof of liveness of the device. The function *setTimeout* lets the owner establish how much time the device has to again prove that it is operating properly. Device malfunctioning can be detected with the function *checkTimeout*. This function is employed in many of the functions described below.

The pseudo-codes of the functions *transferFrom*, and *setUser* are shown in Table 4. The function *transferFrom* defines the new owner and generates the event *Transfer* when finished (if the device is operating correctly). The function *setUser* defines the new user. Only a user can use the device bound to the token. Therefore, if the owner needs to use the device, the owner BCA address has to be defined as user BCA address. If a device is already engaged with an owner, mutual authentication is not required again if the owner is defined as user. The function *setUser* generates the event *UserAssigned* when finished (and the device is operating correctly). If the device is malfunctioning, the event *TimeoutAlarm* is sent. Devices, owners, and users have to be subscribed to events to be aware of the transferences. In the case of devices, these events make them aware of a change in the operating mode, as detailed in the next section.

The pseudo-codes of the functions *startOwnerEngagement*, *startUserEngagement*, *ownerEngagement*, and *userEngagement* are shown in Table 5. The functions *startOwnerEngagement* and startUserEngagement, which are executed by the owner and user, respectively, save the public data *dataEngagement* and the hash of the secret they propose to share. The functions *ownerEngagement* and *userEngagement*, which are executed by the device, check if the device agrees with the secret. If the checking is successful, the token state changes from Waiting for owner to Engaged with owner and from Waiting for user to Engaged with user, respectively.

Given the device BCA address, the functions *tokenFromAddress*, *ownerOfFromAddress* and *userOfFromAddress* return, respectively, the token identifier, the address of owner, and the address of user. Given the tokenId, the function *userOf* returns the address of the user. The tokenIds assigned to a user are returned by the function *userBalanceOf*, and the tokenIds of a particular owner assigned to a user are returned by the function *userBalanceOfAnOwner*.

## 4. Proposal of Physically Binding IoT Devices to Smart NFTs Using PUFs

### 4.1. Creation of Smart NFTs

A blockchain is cryptographically secure because the participants sign their transactions. To be allowed to sign transactions, it is fundamental to have a public/private key pair and an address associated with a blockchain account (BCA). The BCAs are unique and univocally identify the participants. From the private key, the public key and the address are derived. Authenticity, integrity, and non-repudiation are ensured only if the private key is kept secret and is properly protected. If not, an impersonation attack can be carried out since the key pair conforms the identity of the participant. In this work, physical unclonable functions (PUFs) are used to obfuscate and reconstruct the private keys on the fly. Since PUF responses are noisy, a fuzzy extractor or helper data algorithm (HDA) is needed. The fundamental primitive of the HDA is an error correction [*n*, *k*, *d*] code *C* with coding and decoding algorithms capable of correcting up to *t* errors given a distance *d*. An enrollment phase is performed to generate the device private key *SK_DEV_*, on the one hand, and the public Helper Data *HD_DEV_* needed to reconstruct it, on the other hand. In this work, it is assumed that *SK_DEV_* is generated from a random seed provided by a true random number generator (TRNG). *HD_DEV_* is generated by mapping *SK_DEV_* to a codeword *cs* through the coding algorithm of *C* and later by XORing it with a PUF response *R* of length *n*. To reconstruct *SK_DEV_*, *HD_DEV_* is XORed with a fresh PUF response *R’* and, later, it is applied to it the decoding algorithm of *C*. In this form, the private key is not stored anywhere, and only the genuine device can recover it. 

SRAM PUFs are employed in this work since SRAMs can be found in the majority of IoT devices and no additional hardware is required to be included or implemented. Hence, these types of PUFs have a low cost. Basically, an SRAM cell is formed by six transistors, four of which are used as two cross-coupled inverters and two of which are used as access transistors. A write operation can force the output value toward 0 or 1 depending on the applied input. However, if the cell is powered up and no external signal is applied as input, the output value of the cell (known as the start-up value) is fixed by the inverter that begins to conduct. This outcome is unpredictable without having direct access to the cell since it depends on the transistor mismatching determined by random variations of the semiconductor fabrication process. The start-up value of the cell is not necessarily the same between different power-ups due to the presence of noise. This bit flipping is generally unwanted when the SRAM cell start-up value is used in the PUF response, since the PUF reproducibility is reduced and the complexity of the error correction code raises accordingly. To mitigate this, the solution explained in [32] is used herein, which only employs the most stable cells to generate the SRAM PUF response. If the resulting SRAM PUF responses show some bias, a debiasing algorithm has to be considered. The unbiased stable cells after this classification process are named ID cells. Using these ID cells, a simple repetition error correction code works correctly, as shown in [16,32].

Unlike ID cells, which show stability in several power-ups, there are cells that are very unstable by nature and their start-up values vary greatly. These cells can be used as a TRNG, as explained in [33]. In this work, these cells, named as RND cells, are used as a source of entropy for the generation of the IoT device private keys employed for the BCAs and for the generation of the nonces employed in the communication protocols.

As illustrated in Figure 2, prior to create the token, the manufacturer challenges the SRAM PUF of the IoT device. The IoT device carries out the cell classification from which the *ID_MASK_* and *RND_MASK_* are generated. With the application of the *RND_MASK_*, the start-up values of the RND cells are obtained and then employed as input to a hash function that outputs the private key. The device BCA address, *device*, is derived from the application of a public key generator to the private key. With the application of the *ID_MASK_*, the ID cells are obtained. The Helper Data *HD_DEV_* is the result of applying an XOR operation to the start-up values of the ID cells and the codeword associated to the private key. Then, the IoT device sends *ID_MASK_*, *RND_MASK_, HD_DEV_*, and *device* to the manufacturer. The manufacturer creates the smart NFT with the transaction associated to the function *createToken* by indicating the owner and device BCA addresses. Once the smart NFT is created, the manufacturer receives the tokenId and programs the device. 

It is assumed in this work that the SoC (system on a chip) acting as the processing core of the IoT device contains a small one-time programmable (OTP) memory, e.g., a ROM, and a larger non-volatile memory (NVM), e.g., a flash or SD card. The manufacturer writes a very small piece of data and code (the zero-stage bootloader *ZSB*, the manufacturer public key *PK_MAN_,* and the tokenId) in the OTP memory and most of the data and code (the first-stage bootloader *FSB*, the kernel of the operating system *OS_MAN_*, and the firmware needed to interact with the blockchain *FW_BC_*, including the *ID_MASK_*, *RND_MASK_*, and *HD_DEV_*) in the NVM. Since the NVM is not tamper-proof, all the data stored in it are signed with the manufacturer private key *SK_MAN_*. In the OTP memory, the tokenId is stored, since it represents the logical and immutable identity of the smart NFT associated with the device. In addition, the zero-stage bootloader is stored together with the manufacturer public key *PK_MAN_* and the code needed to verify the signatures.

### 4.2. Operating Modes of the Device

An IoT device, programmed and bound to a smart NFT by a manufacturer, executes a thread, herein named *FW_BC_*, which consults in the blockchain its token information and updates the timestamp. Typically, this is scheduled by the operating system kernel *OS_MAN_* provided by the manufacturer. A timeout interruption determines when the device requests and updates the token information from the blockchain through its BCA address. The first value of timeout is given by the manufacturer. Then, the owner can change this value and the device takes it from its smart NFT. The device knows its operating mode from the state of its smart NFT. 

When the smart NFT is created, the operating mode of the device defined in the state is *Waiting for owner*. If the state is *Waiting for owner*, the device saves in its memory the owner BCA address. Since devices, owners, and users agree on the elliptic curve employed (secp256k1 in Ethereum) and the primitive element P used on this curve, the owner generates a pair of keys, a private key ($SK_{OD}$) and a public key ($PK_{OD}$), which verify that $PK_{OD}=SK_{OD}\cdot P$. The owner proposes $K\_O=SK_{OD}\cdot PK_{DEV}$ as secret to share with the device to establish a secure communication channel. The owner publishes $PK_{OD}$ as *dataEngagement* and the hash of K_O as *hashK_O* with the function *startOwnerEngagement*. The owner requests the device to finish engagement. The device validates the owner request, generates its secret as $K\_D=SK_{DEV}\cdot PK_{OD}$, using the public data $PK_{OD}$, and sends the hash of K_D as *hashK_D* with the function *OwnerEngagement*. If everything is correctly done, K_O and K_D are the same since
$${K\_O=SK}_{OD}\cdot {PK}_{DEV}={SK}_{OD}\cdot \left({SK}_{DEV}\cdot P\right){=SK}_{DEV}\cdot \left({SK}_{OD}\cdot P\right){=SK}_{DEV}{\cdot PK}_{OD\;}=K\_D.$$

Hence, the function *OwnerEngagement* changes the state of the token to *Engaged with owner*, updates the timestamp, and sends the event *OwnerEngaged*. Once the device receives the event, it changes its operation mode to *Engaged with owner*. This process is shown in Figure 3. From this moment, the device can be managed by the owner. The owner can configure and program the device according to the application, uploading the software *SW_OWNER_*. Then, the owner can assign a user to the device, without losing the ownership of the device. If the owner sends the transaction *setUser* to the smart contract, this function changes the state of the token to *Waiting for user* and sends the event *UserAssigned*. 

If the device consults the blockchain and the state of its smart NFT is *Waiting for user*, the device saves in its memory the user BCA address. Then, a mutual authentication process is carried out with the user, as already done with the owner. The user generates a pair of keys, a private key ($SK_{UD}$) and a public key ($PK_{UD}$), proposes $K\_U=SK_{UD}\cdot PK_{DEV}$ as secret to share with the device to establish a secure communication channel, and publishes $PK_{UD}$ as *dataEngagement* and the hash of K_U as *hashK_U* with the function *startUserEngagement*. The user requests the device to finish engagement. The device validates the user request, generates its secret as $K\_D=SK_{DEV}\cdot PK_{UD}$, and sends the hash of K_D as *hashK_D* with the function *userEngagement*. If everything is correctly done, K_U and K_D are the same, and the function changes the state of the token to *Engaged with user*, updates the timestamp, and sends the event *UserEngaged*. Once the device receives the event, it changes its operation mode to *Engaged with user*. This process is shown in Figure 4. From this moment, the user can employ the device and upload their own software, *SW_USER_*, to configure and program the device accordingly to its use. If the mutual authentication of the device with the owner or with the user is not finished with engagement, the owner or user can publish the private key ($SK_{OD}$ and $SK_{UD}$, respectively) if they want to show that the fail was on the side of the device and not on their side.

The shared secrets agreed between devices and owners and between devices and users are sensitive information since, from them, and using a key derivation function (KDF), fresh session cryptographic keys can be derived for secure communication. Hence, the device obfuscates them with its PUF and reconstructs them with helper data stored in its NVM, in the same way as explained for its private key, *SK_DEV_*. Once reconstructed, the device checks them with the attributes *hashK_OD* and *hashK_UD* in its token. 

### 4.3. Trusted Software and Hardware of IoT Devices

As commented above, the code to be executed by the IoT device (*OS_MAN_*, *FW_BC_*, *SW_OWNER_*, and *SW_USER_*) usually resides in the NVM. The problem is that attackers can manipulate these external non-volatile memories and, thus, change the behavior of the IoT device. For its proper behavior, all of this code should be trusted. In particular, the owner and user software, *SW_OWNER_* and *SW_USER_*, respectively, as well as the other threads of the operating system kernel *OS_MAN_*, should not be able to interfere with the thread *FW_BC_* used to interact with the blockchain or consult and update the token information. If this is not accomplished, the link between the device and its token can be broken. A way to ensure the permanence of this link is by means of a secure boot, which proves the trustworthiness of the code early at boot time. The code at each boot stage is verified using a digital signature in a chain-of-trust manner. 

In the initial stage, the code called *Zero-Stage Bootloader (ZSB)*, which is located in the internal OTP memory of the main SoC, is executed. Since this code cannot be modified, it is considered the *Root of Trust (RoT)* of the device. We assume a non-leakage chip model [34] for the main SoC. This means that all the internal components of the main SoC are trusted, including the on-chip SRAM, which also acts as an SRAM PUF. It is also assumed that the manufacturer is a trusted entity, which does not introduce any malware or hardware trojan.

We propose to include a secure boot process in the IoT device hardware with the following steps, shown in Figure 5: The *Zero-Stage Bootloader* verifies the *First-Stage Bootloader* (*FSB*) signed by the manufacturer. The signature *SIG(FSB, SK_MAN_)* stored in the external NVM is verified with the manufacturer public key *PK_MAN_* (which is associated with the manufacturer private key *SK_MAN_*). The manufacturer public key is trusted since it is also stored in the OTP memory. Then, if the verification is successful, the *First-Stage Bootloader* is executed.The *FSB* reads the partition table and verifies the integrity of the operating system kernel *OS_MAN_* using the manufacturer public key *PK_MAN_* stored in the internal OTP memory and the signature *SIG(OS_MAN_, SK_MAN_)* stored in the external non-volatile memory. If the verification is successful, the kernel of the operating system takes the control of the IoT device.The kernel *OS_MAN_* launches *FW_BC_*, after verifying it, so that the device can interact with the blockchain using its own BCA address. To do this, the *FW_BC_* reconstructs the private key. The tuple *{ID_MASK_, HD_DEV_}* is read from the external NVM, and the start-up values of the ID cells determined by the *ID_MASK_* are read from the on-chip SRAM. Later, the start-up values are XORed with the *HD_DEV_*, and the repetition error decode is applied. Once the private key is reconstructed, the public key is derived and the IoT device is able to generate its BCA address from it. Using the device BCA address, the device checks if the tokenId saved in its OTP memory is the one associated with its BCA address in its smart NFT. With this checking, the device proves its hardware authenticity. Then, the device retrieves the rest of its NFT attributes, such as its owner and user BCA addresses and its state, and it updates its timestamp. Depending on the NFT state, the device executes the code associated with the *Waiting for owner*, *Engaged with owner*, *Waiting for user*, or *Engaged with user* operating modes. If the device is already engaged with an owner, the signature *SIG(SW_OWNER_, SK_OWNER_)* is verified with the owner public key *PK_OWNER_*, which is related to the owner BCA address. If the verification is successful, the application *SW_OWNER_* is executed. In addition, if the device is already engaged with a user, the user application *SW_USER_* is verified with the retrieved user BCA address and the signature *SIG(SW_USER_, SK_USER_)*. If the verification is successful, the application *SW_USER_* is executed. These applications are executed concurrently to *FW_BC_*, which periodically consults and updates the device token. The shared secrets can be employed to interchange confidential data.

## 5. Proof of Concept with ESP32-Based IoT Devices and Ethereum Blockchain

A proof of concept was developed on the basis of a Pycom Wipy 3.0 board as IoT device (which includes an Espressif ESP32 microcontroller as the main SoC) and the public Ethereum blockchain. ESP-IDF was employed to develop the required device functionalities, Remix was employed to develop the smart contract of the smart NFT, and three DApps (Decentralized Applications) were created for the roles of manufacturer, owner, and user, as detailed below.

### 5.1. Firmware Verification and Blockchain Account Generation from SRAM PUFs

The proposed secure boot process explained in Section 4.3 lasts longer than a non-secure process since hash operations and signature verifications introduce some overhead. We used the verification algorithm with the secp256k1 curve of the micro-ecc library to verify the manufacturer signatures and the SHA512 function to hash the code to be verified, taking advantage of the SHA accelerator included in the ESP32 microcontroller. The time it took to verify a signature was 165.7 ms. The hashing operation that has to be done prior to the signature verification depends, of course, on the code size of the stage to be verified. The *FSB* on the ESP32 can occupy around 15–17 KB [35] and hashing it took around 2.5 ms. On the other hand, the size of the operating system kernel OS_MAN_ can vary depending on the features required by the application context. For example, the Moongose OS operating system [36], with several features for enhanced IoT devices, can occupy at least 113 KB and hashing a code of that size took 19.5 ms. Therefore, the signature verification was the slowest operation in the secure boot process.

In previous works [16,37], the internal SRAM of the ESP32 microcontroller was evaluated as a PUF, considering as the most stable (STB) cells the SRAM cells whose start-up value did not change in 20 consecutive measurements. Table 6 summarizes the experimental results. Three different boards were used, and the experiments were done in nominal operation conditions. Note that the majority of SRAM cells are stable cells that do not show bias. For evaluating PUF uniqueness and reproducibility, the average fractional Hamming distance was used as follows:$${\frac{{dist}_{H}}{n}]}_{avg}=\frac{2}{n\cdot k\cdot (k-1)}\cdot \overset{k-1}{\underset{i=1}{\sum }}\overset{k}{\underset{j=i+1}{\sum }}{dist}_{H}{(R}_{i}{,\;R}_{j}),$$
where $dist_{H}$ is the Hamming distance, n is the number of bits of the responses, and $R_{i}$ and $R_{j}$ are PUF responses. Inter-Hamming distance (from PUF responses from different SRAM cells) was used to evaluate PUF uniqueness, and intra-Hamming distance (from PUF responses from the same SRAM cells) was used for PUF reproducibility. Since the average inter-Hamming distance was nearly 0.5, this confirmed PUF uniqueness. Since the average intra-Hamming distance was very close to 0, this showed that PUF reproducibility was fairly good. It was concluded that a repetition error correcting code of length 8 was sufficient to properly recover a 256-bit obfuscated secret seed. Hence, 2048 ID (STB) cells were needed [16].

The most unstable (or RND) cells of internal SRAM of the ESP32 microcontroller were also studied. Having performed 20 measurements, those cells had a start-up value of “0” half of the time and “1” the other half of the time. Since they provided the minimum entropy shown in Table 6, 369 RND cells were needed to generate a 256-bit random sequence with full entropy (256/0.6952 cells) [37].

After 20 readings of SRAM start-up values, the device generates the *ID_MASK_* and *RND_MASK_*. Although this classification process is slow, as shown in Table 7, it is only executed once previously to use the device and create the token, and this allows accelerating the processes that should be repeated afterwards. The device uses the *RND_MASK_* to obtain a secret seed. Using the secret seed and a hash, the IoT device generates its 32-byte private key. It is encoded with the 8-bit repetition error correction encoder, and the result is XORed with the start-up values of 2048 ID cells to produce the helper data *HD_DEV_*.

Once the *FW_BC_* is uploaded to the device, the helper data are XORed with the start-up values of the SRAM cells identified by the *ID_MASK_*. Using the 8-bit error repetition decoder, the private key is recovered. Since our proof of concept uses the Kovan testnet Ethereum blockchain, the curve secpt256k1 is needed to obtain the 64-byte public key. The Trezor library was used for the curve secpt256k1 (instead of the micro-ecc library used in the boot process). This library was used to improve the computing times obtained with the micro-ecc library (14.08% reductions in the execution times were achieved). This acceleration is possible because the Trezor library saves precomputed data on Flash memory. This option is not possible in the boot process because internal OTP memories tend to be small and the precomputed values require a significant amount of storage. The BCA address is generated as the rightmost 160 bits of the Keccak hash of the public key. The SHA3 library from Trezor library was used for the Keccak hash.

An Ethereum API is required by the Wipy board to interact with the blockchain. In this proof of concept, Infura [38] was employed as the blockchain gateway. In the Infura free version, the iterations with blockchain are limited to 100,000 requests per day, which is more than enough for our proof of concept. To use Infura, the gateway should be registered and configured. Once the Infura project is created, a project ID and a project secret are generated. Selecting the endpoint as Kovan Ethereum testnet, the URL of the API was generated using HTTPS. Figure 6 shows the configuration of our proof of concept and the Wipy board. For security reasons, we modified the project secret and the project ID.

The interaction between the Wipy board and Infura should employ JSON (Java Script Object Notation) format in the messages. Although this format is a standard derived from JavaScript exchange messages to minimize data size, it is currently used in any language indistinctly. In the case of the Wipy boards, the cJSON library was employed to construct JSON structures in C language. The interaction with the blockchain or smart contract should use a remote procedure call (RPC) protocol encoded in JSON (JSON-RPC). All the available methods in Ethereum using JSON-RPC from Infura can be found in [39]. We took the proper methods from this source.

Table 7 shows the execution times obtained with the Wipy board, concerning private key obfuscation and reconstruction, BCA address generation, and blockchain transaction.

The transaction completion time, measured as the total time from the private key reconstruction to the transfer of a message to the blockchain API, can be short enough for many applications.

### 5.2. The Smart NFT-Based Smart Contract

The smart NFT-based smart contract was developed in Solidity language on Remix IDE [18]. Remix is an official IDE for Ethereum blockchain and has all the assets needed to code, compile, test, and deploy a smart contract. The code developed is available publicly on GitHub [40] and can be deployed on the Kovan testnet with the address 0x7eB5A03E7ED70ABf70fee48965D0411d37F335aC.

Saving a transaction on a block of Ethereum has a cost. It depends on the priority and size of the transaction and the computational cost of the function. In the smart contract proposed, the functions *createToken*, *transferFrom*, *startOwnerEngagement*, *ownerEngagement*, *setUser*, *startUserEngagement*, *userEngagement*, *setTimeout*, *checkTimeout*, and *updateTimestamp* have transaction costs. The gas consumption of these functions, at the time of writing this paper, is shown in Table 8. The costliest function is *createToken* because it saves all the information to generate a token. Comparisons to other proposals in the literature which employ NFTs are not included since those smart contract functions are oriented to specific applications such as accesses to resources and traceability of products. Therefore, gas consumption results of those functions are not comparable.

### 5.3. DApp Interfaces

In addition to developing the smart contract with Remix and the required device functionalities with ESP-IDF, DApp (Decentralized Application) interfaces were created for the roles of manufacturer, owner, and user to have a whole demonstration. On the one hand, these DApps have a connection with the blockchain using a web interface and Metamask [19]. On the other hand, they have a physical connection with the device using UART (Universal Asynchronous Transmitter Receiver) serial communication. UART was selected for simplicity, but other connections such as Bluetooth or WiFi can be employed. This is shown in Figure 7.

While Infura was used as the gateway between the blockchain and the IoT device, Metamask was used in the DApps as the gateway between the blockchain and the web client employed by the manufacturer, owners, and users. Metamask is a browser extension that lets one sign and interact with the blockchain using a key vault of blockchain accounts. This software can be considered as an Ethereum wallet and Ethereum API. Metamask can save one or more blockchain accounts, but only one can be active each time. Hence, it is important to have activated the correct BCA (manufacturer, owner, or user) when using the DApps. In this work, we selected the Kovan testnet to work with Metamask. Since Metamask can be used only with a web client, a Node.js web server was developed [41]. We used the JavaScript runtime environment Node.js due to its simplicity. In order to use Metamask with a web client, the Web3 library together with a Metamask Legacy Web3 browser extension was employed for the DApps.

The middleware between the DApp web client and the IoT Device was also designed with Node.js. It connects the UART with the WebSocket protocol, which uses a Transmission Control Protocol (TCP). When the DApp web client sends a WebSocket message to the middleware, the header of the message indicates what the middleware should do, such as requesting the helper data from the IoT device or requesting the hashK_D, among other functions. On the other hand, when the DApp web client receives a WebSocket message, it also executes the proper function depending on the message header.

The manufacturer DApp is the one used by the manufacturer to upload, in the IoT device, the firmware needed for the device registration process. This firmware carries out the creation of the *ID_MASK_* and *RND_MASK_*, the helper data *HD_DEV_*, and the BCA of the device. The DApp creates a UART connection with the IoT Device to obtain this information. In addition, in the token creation process of the smart contract, the DApp obtains the *tokenId* and registers the token in the blockchain with an owner. With all this information, the manufacturer DApp creates the firmware *FW_BC_* of the device and uploads the *OS_MAN_* with the *FW_BC_* into the device.

The owner DApp is used for the mutual authentication between the owner and the IoT device. The DApp requests the hashK_D from the device, uses Metamask to sign this request with the active BCA, and sends $PK_{OD}$ to the device. The DApp makes the smart contract execute the function *startOwnerEngagement* to start the engagement. Once the owner is engaged (the event OwnerEngaged is received), this DApp allows uploading the SW_OWNER_ into the device and the assignation of users by the owner.

The user DApp is used for the mutual authentication between the user and the IoT device, similarly to the mutual authentication with the owner DApp. This DApp makes the smart contract execute the *startUserEngagement* function to start the engagement. Once the user is engaged (the event UserEngaged is received), this DApp allows uploading the SW_USER_ into the device. Figure 8 illustrates these DApp interfaces.

### 5.4. Example of Use Case

The proposed solution can be very efficient in many scenarios. As an illustrative example, let us consider a certification company in charge of acquiring measurements (such as levels of noise, radioactivity, or carbon dioxide emissions) that should achieve regulation compliance to prevent harm to the environment, citizens, and industry workers. Trusted IoT devices should be used to obtain trusted measurements. The employment of the proposed smart NFTs in this use case contributes to the security of the IoT device from manufacturing to operation. The IoT device cannot be replaced by a counterfeit one and its software cannot be manipulated if it is bound to a smart NFT. The certification company, acting as owner of the devices, can set inspectors as users. Thanks to the mutual authentication processes, which are registered in the smart NFT, session keys can be established to transmit confidential measurements taken by the IoT device to the certification company or to the inspector. On the other hand, the certification company and the inspector can load and execute their applications related to measurement setup or acquisition. The secure boot process checks if the software (from the manufacturer, the certification company, or the inspector) is the expected one. The IoT device can be transferred to another certification company or inspector, and the changes are registered in the smart NFT. The measurements performed by the IoT device can be hashed and uploaded in the blockchain together with the smart NFT information to provide a complete traceability of the certification procedure.

## 6. Conclusions

The solution proposed in this work guarantees the trustworthiness of the hardware and software of IoT devices from manufacturing until end-user application. The proposed smart non-fungible tokens (NFTs) extend the attributes defined by the ERC-721 standard (token identifier and owner BCA address) to also include the device and user BCA addresses, the states associated with the operating modes of the device, public data related to secrets shared with its owner and user, and time information about the last link checked between the token and the device. The gas consumption of the smart contract functions defined for the token (developed in Solidity) is given. The function to create the token is the costliest because it saves all the information to generate the token, while the gas consumption of the smart contract functions defined for checking and updating the link with the device is much smaller.

The IoT device is able to generate and recover its BCA in a trusted way from its hardware through a PUF and executes a secure boot that ensures it consults its smart NFT periodically. In this way, the device is physically bound to its Smart NFT because the token identifier stored in its one-time programmable memory matches the device BCA address recovered and stored in the token. In addition, secure communication channels can be established between devices, owners, and users. In the proof of concept developed with a Wipy 3.0 board (which includes an ESP32 microcontroller), the internal SRAM of the ESP32 microcontroller was employed as a PUF. The time needed to complete a transaction was 52.65 ms, considering the recovering of the 32-byte private key, the 64-byte public key regeneration, the 20-byte BCA address for Ethereum, the message preparation, and the transfer to Infura blockchain API. Concerning the secure boot process of the device (which is based on hash operations and signature verifications), the time is dominated by the signature verification (the ESP32 microcontroller has a SHA accelerator). The time to verify a signature was 165.7 ms (using secp256k1 curve and the micro-ecc library). The time needed to generate a shared secret was 166.6 ms (using the Trezor library). These times are good for many applications. In order to complete the demonstrator, DApp (Decentralized Application) interfaces were created for the roles of manufacturer, owner, and user of the devices.

## Figures and Tables

**Figure 1 sensors-21-03119-f001:**
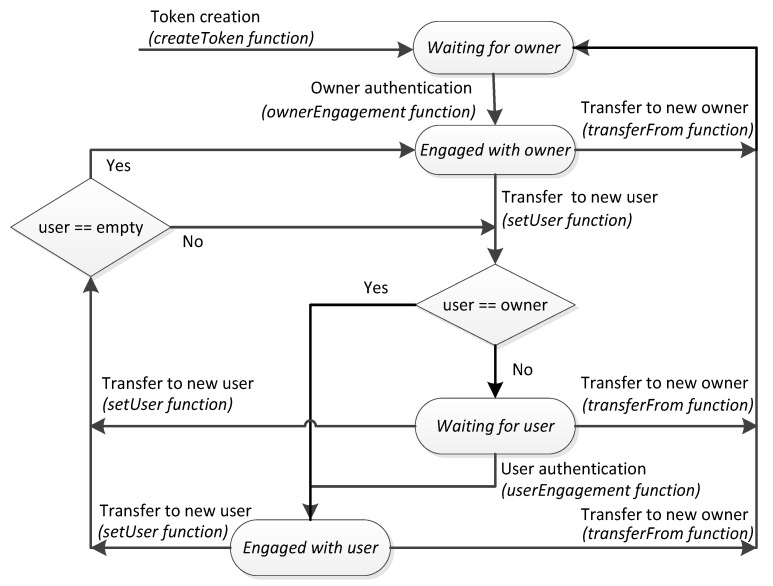
State diagram of smart NFTs.

**Figure 2 sensors-21-03119-f002:**
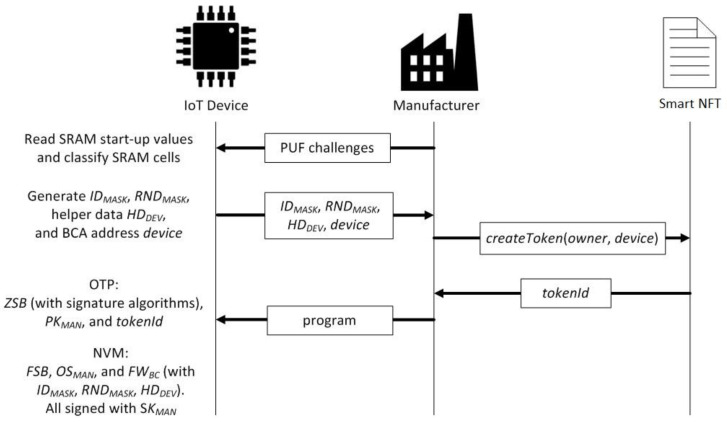
The manufacturer binds the IoT device to its smart NFT.

**Figure 3 sensors-21-03119-f003:**
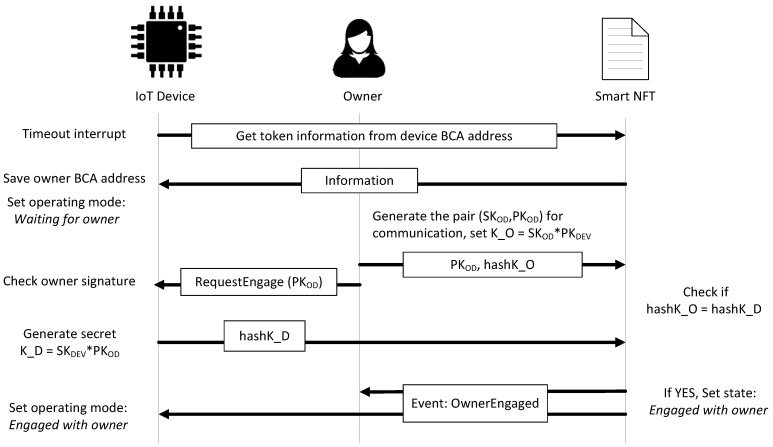
Steps in successful owner and device mutual authentication.

**Figure 4 sensors-21-03119-f004:**
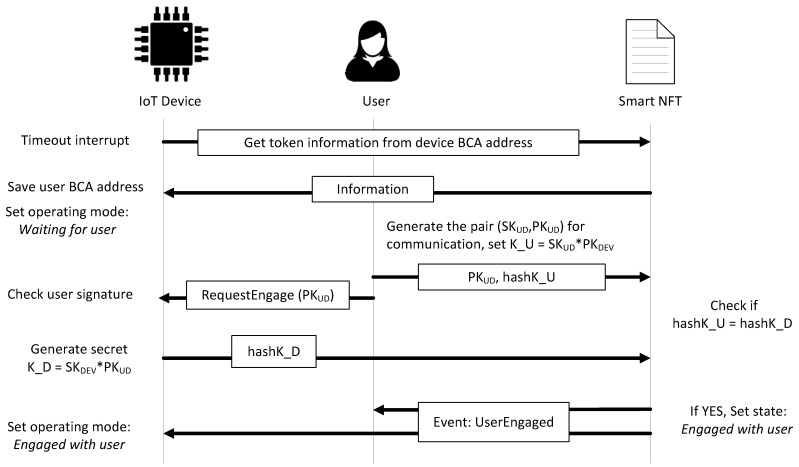
Steps in a successful user and device mutual authentication.

**Figure 5 sensors-21-03119-f005:**
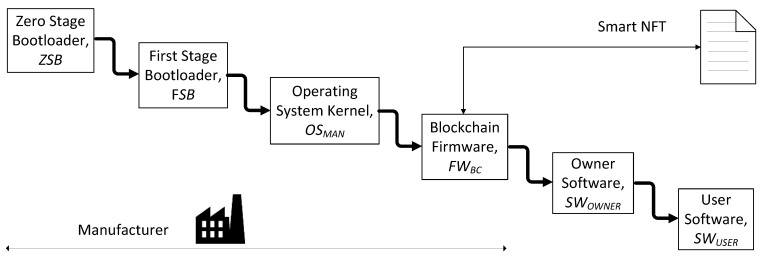
Proposed secure boot process of the IoT device.

**Figure 6 sensors-21-03119-f006:**
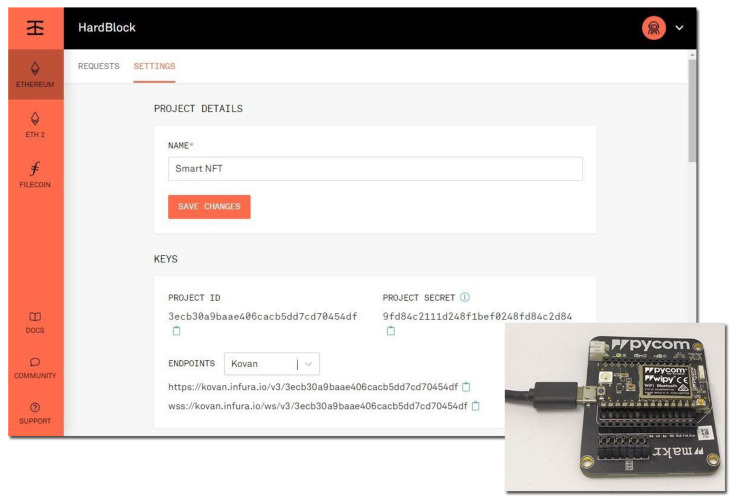
Configuration of our proof of concept in Infura. The Pycom Wipy 3.0 board is also shown.

**Figure 7 sensors-21-03119-f007:**
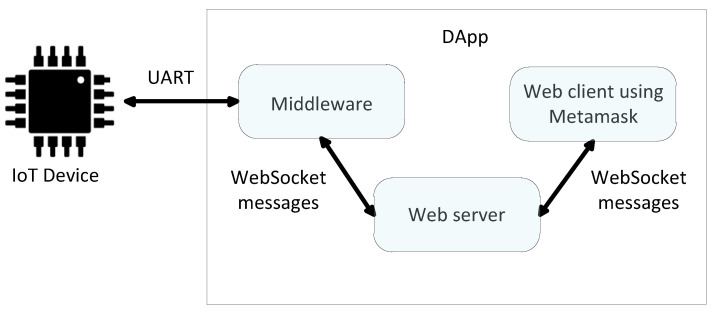
Structure of the communication between the IoT device and the DApps.

**Figure 8 sensors-21-03119-f008:**
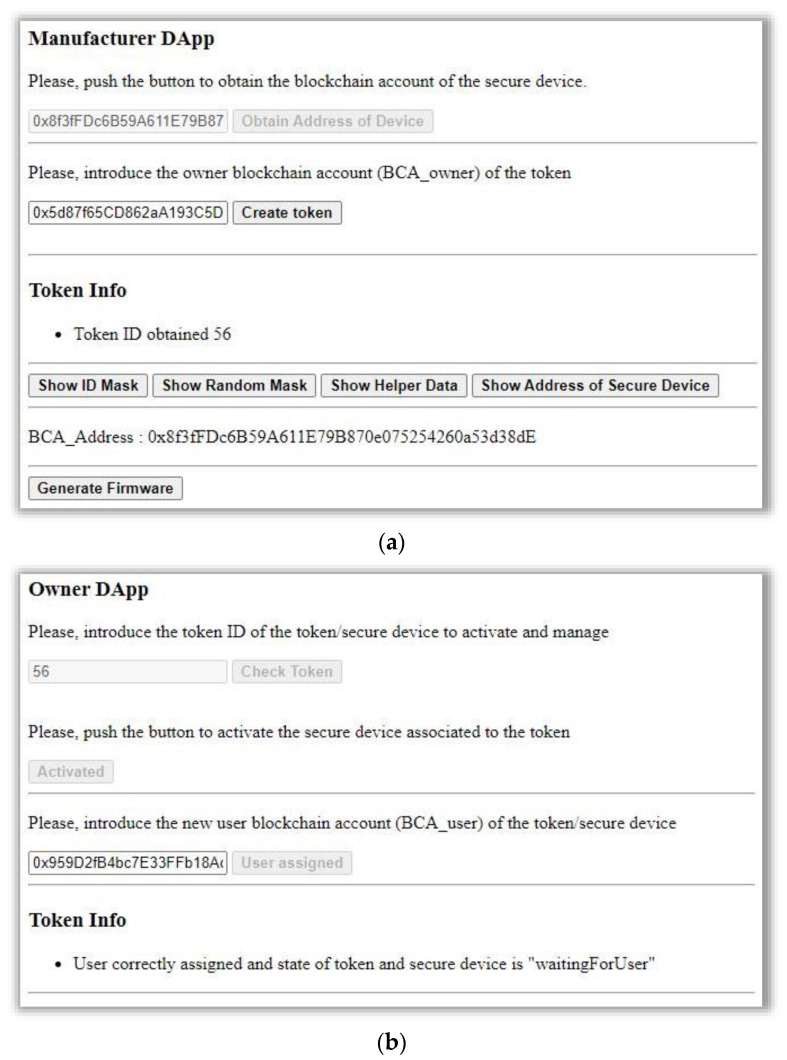
(**a**) Manufacturer DApp, (**b**) owner DApp, and (**c**) user DApp.

**Table 1 sensors-21-03119-t001:** Attributes of smart NFTs.

Type	Name of Variable	Defined by ERC-721
uint256	*tokenId*	Yes
address	*owner*	Yes
address	*device*	No
address	*user*	No
enum	*state*	No
uint256	*hashK_OD*	No
uint256	*hashK_UD*	No
uint256	*dataEngagement*	No
uint256	*timestamp*	No
uint256	*timeout*	No

**Table 2 sensors-21-03119-t002:** Functions and events of Smart NFTs.

Functions and Events	Defined by ERC-721
function *transferFrom* (address _from, address _to, uint256 _tokenId) external payable;	Yes
function *ownerOf* (uint256 _tokenId) external view returns(address);	Yes
function *balanceOf* (address _owner) external view returns(uint256);	Yes
function *createToken* (address _device, address _owner) external view returns (uint256);	No
function *startOwnerEngagement* (uint256 _tokenId, uint256 _dataEngagement, uint256 _hashK_O) external;	No
function *ownerEngagement* (uint256 _tokenId, uint256 _hashK_D) external;	No
function *startUserEngagement* (uint256 _tokenId, uint256 _dataEngagement, uint 256 _hashK_U) external;	No
function *userEngagement* (uint256 _tokenId, uint256 _ hashK_D) external;	No
function *setUser* (uint256 tokenId, address _user) external;	No
function *tokenFromAddress* (address _device) external view returns (uint256);	No
function *ownerOfFromAddress* (address _device) external view returns (address);	No
function *userOfFromAddress* (address _device) external view returns (address);	No
function *userOf* (uint256 _tokenId) external view returns (address);	No
function *userBalanceOf* (address _user) external view returns (address);	No
function *userBalanceOfAnOwner* (address _user, address _owner) external view returns (address);	No
function *updateTimestamp* () external;	No
function *setTimeout* (uint256 _tokenId, uint256 _timeout) external;	No
function *checkTimeout* (uint256 _tokenId) external returns (bool);	No
event *Transfer* (address _from, address _to, uint256 _tokenId);	Yes
event *OwnerEngaged* (uint256 _tokenId);	No
event *UserAssigned* (uint256 _tokenId, address _user);	No
event *UserEngaged* (uint256 _tokenId);	No
event *TimeoutAlarm* (uint256 _tokenId);	No

**Table 3 sensors-21-03119-t003:** Pseudo-codes of the functions related to the creation and configuration of smart NFTs.

*createToken*: Creates a New Token Linking a Device BCA Address to a TokenId
Input: _device, _owner
Output: tokenId
Require manufacturer = = msg.sender
Generate new tokenId
Set owner of tokenId = _owner
Set state of tokenId = *Waiting for owner*
Set BCA address of tokenId = _device
Set timestamp = block timestamp
Return tokenId
*updateTimestamp*: The device updates the attribute timestamp of its token
Set timestamp of tokenFromBCA(msg.sender) = block timestamp
*setTimeout:* The owner of the token sets its attribute timeout
Input: _tokenId, _timeout
Require owner of _tokenId = = msg.sender
Set timeout of _tokenId = _timeout
*checkTimeout:* Checks if the device remains bound to its token
Input: _tokenId
if timestamp + timeout < block timestamp
then return false
else return true

**Table 4 sensors-21-03119-t004:** Pseudo-codes of the functions of smart NFTs related to the definitions of new owners and users.

*TransferFrom*: Transfers a Token from an Owner to a New Owner
Input: _old_Owner, _new_Owner, _tokenId
Require {owner, operator, approved} of _tokenId = = msg.sender
if checkTimeout = = true
then Set user BCA address from _tokenId = _user
and Set {dataEngagement, hashK_UD} = 0
and if _user = = 0 then Set state of _tokenId = *Engaged with owner*
and else Set state of _tokenId = *Waiting for user*
and Send event UserAssigned
else Send event TimeoutAlarm
*setUser: The owner assigns a user to the token*
Input: _tokenId, _user
Require state is *Engaged with owner*, *Waiting for user or Engaged with user and owner* of _tokenId = = msg.sender
if checkTimeout = = true
then Set user BCA address from _tokenId = _user
and Set {dataEngagement, hashK_UD} = 0
and if _user = = 0 then Set state of _tokenId = *Engaged with owner*
and else Set state of _tokenId = *Waiting for user*
and Send event UserAssigned
else Send event TimeoutAlarm

**Table 6 sensors-21-03119-t006:** Summary of experimental results of SRAM PUFs and TRNGs in ESP32 [16,37].

Experiment	Result
No. SRAM cells evaluated per board	237,320
No. STB cells (min.)	199,514
Bias in STB cells	No
No. SRAM cells per PUF response	2048
No. PUF responses	291 (3 × 97)
Avg. intra fract. Hamming dist.	0.0025
Avg. inter fract. Hamming dist.	0.4921
No. RND cells	724
Min. entropy (%)	69.52

**Table 7 sensors-21-03119-t007:** Execution times of the main operations performed by the IoT device.

Operation		Execution Time (ms)
Private key obfuscation	SRAM cells classification	3.8 × 10^5^
	32-byte private key generation (RND mask and hash application)	0.41
	ID mask application, repetition error correction code and XOR operation	2.02
Shared secret generation	Product of the private and public keys	166.6
Zero-stage boot	Hashing 15–17 KB	2.5
	Signature verification	165.7
Private key reconstruction	ID mask application, XOR operation, and repetition error correction code	1.60
BCA address generation	64-byte public key generation (secp256k1 operation)	21.15
	20-byte BCA address creation (Keccak256 operation)	0.45
Blockchain transaction	Message preparation (creation of JSON structure and signature)	26.10
	Transfer to Infura blockchain API	2.90
Transaction completion		52.65

**Table 8 sensors-21-03119-t008:** Gas consumption of the main functions implemented.

Function	Gas Consumption
createToken()	167,263
transferFrom()	64,556
startOwnerEngagement()	69,216
ownerEngagement()	47,962
setUser()	66,788
startUserEngagement()	69,990
userEngagement()	32,983
setTimeout()	28,874
checkTimeout()	26,429
updateTimestamp()	28,162

**Table 5 sensors-21-03119-t005:** Pseudo-codes of the functions of smart NFTs related to the engagement with owners and users.

*startOwnerEngagement*: Starts Engagement Process between Owner and Device
Input: _tokenId, _dataEngagement, _hashK_O
Require state is *Waiting for owner* and owner of _tokenId = = msg.sender
if checkTimeout = = true
then Set dataEngagement of _tokenId = _dataEngagement and Save _hashK_O
else Send event TimeoutAlarm
*ownerEngagement*: Notifies owner and device their mutual authentication
Input: _tokenId, _hashK_D
Require state is *Waiting for owner* and dataEngagement is not zero and BCA address of _tokenId = = msg.sender
if _hashK_D = = _hashK_O
then Set state of _tokenId = *Engaged with owner*
and Set hashK_OD of _tokenId = _hashK_O
and Set timestamp = block timestamp
and Send event OwnerEngaged
*startUserEngagement*: Starts engagement process between user and device
Input: _tokenId, _dataEngagement, _hashK_U
Require state is *Waiting for user* and user of _tokenId = = msg.sender
if checkTimeout = = true
then Set dataEngagement of _tokenId = _dataEngagement and Save _hashK_U
else Send event TimeoutAlarm
*userEngagement*: Notifies user and device their mutual authentication
Input: _tokenId, _hashK_D
Require state is *Waiting for user* and dataEngagement is not zero and BCA address of _tokenId = = msg.sender
if _hashK_D = = _hashK_U
then Set state of _tokenId = *Engaged with user*
and Set hashK_UD of _tokenId = _hashK_U
and Set timestamp = block timestamp
and Send event UserEngaged

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
