# Peer review of "Secure Combination of IoT and Blockchain by Physically Binding IoT Devices to Smart Non-Fungible Tokens Using PUFs"

_sensors, 2021, doi:10.3390/s21093119_

Round 1

Reviewer 1 Report

The article presents a solution based in Smart Non-Fungible Tokens (NFTs) to represent IoT devices. The paper is well-written, though it is clearly based in its reference [14]

[14] Arcenegui, J.; Arjona, R.; Baturone, I. Secure Management of IoT Devices Based on Blockchain Non-fungible Tokens and Phys-639 ical Unclonable Functions. International Conference on Applied Cryptography and Network Security. Springer, Cham, 2020. pp. 24-40. 640 https://doi.org/10.1007/978-3-030-61638-0_2

Major issues
------------

1. As said above, the paper is clearly based in [14]. Though the authors try to explain their new contributions from line 106, I am not really sure if the novel contributions are enough to justify a new paper on this topic.

2. The authors state that "token states are included to represent the operating modes of the IoT device, as it strenghts the security of the operating modes". In fact, only one more attribute is added from their previous work, and it is not clear how deeply it affects to security enforcement. 

3. Attributes and functions are apparently very similar to the ones described in [14]. Moreover, there is no novelty in the use of PUFs, and the processes of binding and authentication have a great similarity with [14].

4. Proof of concept has included some novelties, like the Smart NFT-based Smart Contract and DApp Interfaces, but Table 5 shows exactly the same results as in the authors' previous works.

5. In conclusion, it is a very interesting work, but the authors must justify better that a new article is neccessary

Author Response

First of all, we would like to thank all the interesting comments from the reviewers. As we detail in the following, we hope to have fulfilled all their concerns and suggestions. Modifications in the new version of the paper have been indicated.

Reviewer#1, Concern # 1: The paper is clearly based in [14]. Though the authors try to explain their new contributions from line 106, I am not really sure if the novel contributions are enough to justify a new paper on this topic.

Author response: Thanks to the reviewer concern, we have realized that the differences with our previous work were not clearly highlighted. Now, we have added more material to justify a new paper on this topic.

Author action: We updated the manuscript Introduction by remarking the weaknesses of our previous work in [16] (previously, [14]) with the following text: “The binding of ERC-721 NFTs with physical IoT devices using PUFs was proposed in [16]. That work includes, as a new attribute in ERC-721 NFTs, the blockchain account (BCA) address associated with the IoT device. The use of PUFs in the hardware of the IoT device is proposed to reconstruct the private key from which the BCA address of the device in its associated token is derived. In addition, that work also includes, as a new attribute in ERC-721 NFTs, the blockchain account (BCA) address associated with the user of the IoT device, so that not only the ownership but also the use of the device can be traced by the blockchain with the same token. However, that work does not allow detecting if the device is not operating correctly, or if the link between the device and the NFT is broken, or the engagement with owner and user is lost at some time. To avoid those security flaws, this paper introduces a new NFT, named Smart NFT, which strengthens the link between the IoT device and the NFT used to represent it.”.

The main contributions were updated in this manner:

  • The proposal of Smart NFTs to represent IoT devices that can participate actively in the blockchain by its BCA (to receive and provide information, and to sign transactions) and that can operate in several modes. The attributes defined for Smart NFTs are not only the owner BCA address (as in the ERC-721 standard), the user BCA address, and the device BCA address (as in [16]). In addition, token states are included to trace the operating modes of the IoT device, public data are added to trace if secure communication channels are established between a device and its owner and user, as well as timestamps and timeouts are added to register the last validation of the link between the token and the device.
  • A solution that allows IoT devices proving during their lifetime that their hardware and software are trusted because they are able to be bound to their Smart NFTs and operate as expected. Physical attacks, which are common at IoT devices, like their re-placement by counterfeit devices or modifications of the content of their non-volatile memory, are detected after a controlled delay time thanks to the use of PUFs and the execution of a secure boot process.
  • A solution that allows registering in the blockchain whenever shared secrets are agreed between devices and owners, and between devices and users. From them, they can derive fresh session cryptographic keys for secure communication. Therefore, the trustworthiness of the devices can be traced even if they change of owners and users that manage them.”

In addition, we updated the manuscript Related Work by including the following text: “The extension of the ERC-721 token proposed in [14] is not enough to detect when and why the link between a device and its token is broken. In addition, the transference of the token to a new owner is done automatically without registering the smart contract any mutual authentication between the new owner and device. Hence, malicious owners that transfer non-operative devices cannot be proven. Finally, the transference of the token to a new user is completed with the approval of the user but without registering the smart contract any mutual authentication between the new user and device. Hence, malicious users that misuse devices cannot be proven. To represent IoT devices more securely, the following section describes our proposal of Smart Non-Fungible Tokens.”.

Reviewer#1, Concern # 2: The authors state that "token states are included to represent the operating modes of the IoT device, as it strenghts the security of the operating modes". In fact, only one more attribute is added from their previous work, and it is not clear how deeply it affects to security enforcement.

Author response: This is an interesting concern we tackle in this new version. In addition to the token states, which are included as a Smart NFT attribute to represent the operating modes of the IoT device, public data are also included as attributes to trace if secure communication channels are established between a device and its owner and user, as well as, timestamps and timeouts to register the last validation of the link between the token and the device.

Author action: We extended the Smart NFT proposal in Section 3 with the timestamp, timeout, state, hashK_OD, hashK_UD, and dataEngagement attributes: “The first of them, timestamp, registers in the blockchain whenever the device checks it is bound with its token. This is very important to register if the link is alive or not. The second of them, timeout, is the maximum delay time established for the device to prove again the bounding. If it is exceeded, the device is considered to be malfunctioning…”. “The engagements of the device with an owner and with a user are carried out after mutual authentication protocols based on elliptic curve Diffie-Hellman key exchange protocols. These protocols allow a key agreement between the device and its owner, in the one side, and the device and its user, in the other side. Since the establishment of a shared secret is very important for a secure communication between them, we propose the inclusion of the attributes hashK_OD, hashK_UD, and dataEngagement. The first two attributes define, respectively, the hash of the secret shared between the device and its owner and between the device and its user. Devices, owners, and users should check they are using the correct shared secrets. The attribute dataEngagement defines the public data needed for the agreement. If the mutual authentication fails, dataEngagement allows detecting which parts failed. This is more detailed in the following section.”.

In addition, the Smart NFT was extended with the event TimeoutAlarm to notify if the device is malfunctioning and the functions updateTimestamp, setTimeout, checkTimeout, startOwnerEngagement, startUserEngagement, ownerEngagement and userEngagement: “The function updateTimestamp, which can be carried out only by the device, can be seen as a proof of liveness of the device. The function setTimeout lets the owner establish how much time has the device to again prove it is operating properly. Device malfunctioning can be detected with the function checkTimeout.”. “The functions startOwnerEngagement and startUserEngagement, which are executed by the owner and user, respectively, save the public data dataEngagement and the hash of the secret they propose to share. The functions ownerEngagement and userEngagement, which are executed by the device, check if the device agrees with the secret. If the checking is successful, the token state changes from Waiting for owner to Engaged with owner, and from Waiting for user to Engaged with user, respectively.”.

Table 1 was modified with the new attributes. Table 2 was modified with the new event and functions. The pseudocodes of these functions were included in Tables 3 and 5. Since the operating modes of the device depend on these new attributes, functions and events as explained above, Section 4.2 was updated, as well as, Figures 3 and 4. Furthermore, gas consumption in Table 8 was modified according to these functions.

Reviewer#1, Concern # 3: Attributes and functions are apparently very similar to the ones described in [14]. Moreover, there is no novelty in the use of PUFs, and the processes of binding and authentication have a great similarity with [14].

Author response: Thanks to the reviewer concern, we have improved our proposal of Smart NFT. In this new version, the device employs its BCA address, not only to consult information and receive events from the blockchain as in the previous work, but also to interact to its Smart NFT by participating in the processes of binding and authentication. Regarding the use of PUFs, in this new version, PUFs are also considered to obfuscate the shared secrets between devices and owners, and between devices and users (in addition to generate the device BCA address).

Author action: More attributes and functions were added as explained in the concern # 2. Regarding the use of PUFs, the following text was included in Section 4.2: “The shared secrets agreed between devices and owners, and between devices and users are sensitive information since, from them and using a Key Derivation Function (KDF), fresh session cryptographic keys can be derived for secure communication. Hence, the device obfuscates them with its PUF and reconstruct them with Helper Data stored in its NVM, in the same way as explained for its private key, SKDEV. Once reconstructed, the device checks them with the attributes hashK_OD and hashK_UD in its token.”.

Reviewer#1, Concern # 4: Proof of concept has included some novelties, like the Smart NFT-based Smart Contract and DApp Interfaces, but Table 5 shows exactly the same results as in the authors' previous works.

Author response: Thanks to the reviewer concern, we have improved the proof of concept.

Author action: Execution times for shared key generation and zero stage bootloader were added in Table 7. Gas consumption in Table 8 (previously, Table 5) was modified with the new functions. In addition, an example of use case was included in Section 5.4.

Reviewer#1, Concern # 5: In conclusion, it is a very interesting work, but the authors must justify better that a new article is neccessary.

Author response: All the reviewer concerns have contributed to improve the work. We hope the updates included in the new version justify that a new article is necessary.

Author action: The previous work was extended to create a Smart NFT by which the IoT device participates actively in the binding and authentication processes and executes a secure boot that ensures it consults its Smart NFT periodically. In addition, secure communication channels can be established between devices, owners, and users.

Reviewer 2 Report

In this manuscript, the authors propose a novel Smart Non-Fungible Token, implemented in Ethereum, to represent IoT devices in the blockchain in a secure way.

This paper is very well written, well presented, interesting and scientifically (and technically) sound. Personally, I found it very interesting and clear. There are some typos (e.g.: line 491 -> Etherereum), so a careful re-reading is necessary.

The main limitation I see is the absence of a concrete use-case (or a series of use-cases) that can demonstrate the real validity of the solution, even in comparison with other similar existing solutions. However, given the level of detail of the proposal, as well as the presence of a very effective proof-of-concept, I believe that this work can be recommended for publication in this prestigious Journal.

As a single final note, the authors should further explore the literature on protocols for secure smart contract execution on the blockchain, by at least mentioning some recent work on this subject, e.g.:

  • [https://doi.org/10.3390/computation8030067]
  • [https://doi.org/10.3390/electronics9020255]

Author Response

Response to reviewer 2

First of all, we would like to thank all the interesting comments from the reviewers. As we detail in the following, we hope to have fulfilled all their concerns and suggestions. Modifications in the new version of the paper have been indicated.

Reviewer#2, Concern # 1: There are some typos (e.g.: line 491 -> Etherereum), so a careful re-reading is necessary.

Author response: Thanks to the reviewer for indicating us the typos.

Author action: A careful reading was performed and we corrected the typos.

Reviewer#2, Concern # 2: The main limitation I see is the absence of a concrete use-case (or a series of use-cases) that can demonstrate the real validity of the solution, even in comparison with other similar existing solutions.

Author response: Thanks to the reviewer concern we improved the proof of concept.

Author action: The Section 5.4 Example of use case was added with the following text: “The proposed solution can be very efficient in many scenarios. As an illustrative example, let us consider a certification company in charge of acquiring measurements (such as levels of noise, radioactivity, or carbon-dioxide emissions) that should achieve regulation compliance to prevent harm to the environment, citizens, and industry workers. Trusted IoT devices should be used to obtain trusted measurements. The employment of the proposed Smart NFTs in this use case contributes to the security of the IoT device from manufacturing to operation. The IoT device cannot be replaced by a counterfeit one and its software cannot be manipulated if it is bound to a Smart NFT. The certification company, acting as owner of the devices, can set inspectors as users. Thanks to the mutual authentication processes, which are registered in the Smart NFT, session keys can be established to transmit confidential measurements taken by the IoT device to the certification company or to the inspector. In the other side, the certification company and the inspector can load and execute their applications related to measurement setup or acquisition. The secure boot process checks if the software (from the manufacturer, the certification company or the inspector) are the expected one. The IoT device can be transferred to another certification company or inspector, and the changes are registered in the Smart NFT. The measurements performed by the IoT device can be hashed and uploaded in the blockchain together with the Smart NFT information to provide a complete traceability of the certification procedure.”.

In addition, regarding the comparisons in Table 8, we included the following text: “Comparisons to other proposals in the literature which employ NFTs are not included since those smart contract functions are oriented to specific applications such as accesses to resources and traceability of products. Therefore, gas consumption results of those functions are not comparable.”.

Reviewer#2, Concern # 3: As a single final note, the authors should further explore the literature on protocols for secure smart contract execution on the blockchain, by at least mentioning some recent work on this subject, e.g.:

  • [https://doi.org/10.3390/computation8030067]
  • [https://doi.org/10.3390/electronics9020255].

Author response: These recent works are very interesting to be considered in Introduction when smart contracts are mentioned.

Author action: The Introduction was updated by including these works as references [4] and [5], respectively, with the following text: “These scripts are validated as part of transactions by the consensus algorithm and, thus, once validated, the smart contracts are tamper-proof like the other data registered in the blockchain [4]. In addition, protocols are applied to achieve secure execution of smart contracts [5].”.

Round 2

Reviewer 1 Report

The authors have successfully and clearly addressed all my comments, after a hard and thorough work, so I must only congratulate them for their paper.